# Genomic Analysis of Carbapenem-Resistant *Acinetobacter baumannii* Isolated from Bloodstream Infections in South Korea

**DOI:** 10.3390/antibiotics13121124

**Published:** 2024-11-23

**Authors:** Wook Jong Jeon, Yoo Jung Kim, Ju Hui Seo, Jung Sik Yoo, Dong Chan Moon

**Affiliations:** Division of Antimicrobial Resistance Research, National Institute of Health, Korea Disease Control and Prevention Agency, 187 Osongsaengmyeong 2-ro, Osong-eup, Heungdeok-gu, Cheongju-si 28159, Republic of Korea; finalfate@korea.kr (W.J.J.); yyjj0203@korea.kr (Y.J.K.); hi0327@korea.kr (J.H.S.); jungsiku@korea.kr (J.S.Y.)

**Keywords:** *Acinetobacter baumannii*, carbapenem-resistant, bloodstream, genomics

## Abstract

Background/Objectives: Bloodstream infection by carbapenem-resistant *Acinetobacter baumannii* (CRAB) is a serious clinical problem worldwide. To study its clonal relationship and genetic features, we report the draft genome sequence of CRAB strains isolated from human blood in South Korea. Methods: Among *A. baumannii* strains isolated from patients at nine general hospitals in 2020, 12 CRAB strains of different genotypes were selected. Genomic DNA was sequenced using a combination of Illumina MiSeq and Oxford Nanopore MinION platforms. Antimicrobial susceptibility testing was performed using the disk diffusion method. Antimicrobial resistance and virulence genes were investigated in silico using the Center for Genomic Epidemiology server and the Virulence Factors Database. Results: The multilocus sequence types of isolates included ST191, ST195, ST357, ST369, ST451, ST469, ST491, ST784, ST862, ST1933, ST2929, and a novel type, ST3326. The predominant sequence type, ST191, demonstrated close genetic relationships with several isolates, including ST469, ST369, ST195, ST784, ST491, and ST3326, with ST3326 classified as a subgroup of ST191. We found 18 antimicrobial resistance genes and one quaternary ammonium compound resistance gene. All examined strains harbored *bla*_OXA-23_, which is associated with carbapenem resistance. While variations in antibiotic and disinfectant resistance genes were observed, all isolates exhibited similar virulence factors, with the exception of the biofilm and capsule production genes. Conclusions: This nationwide report of the draft genome sequence of patient-derived strains provides valuable insights into the genomic features associated with clonal relationships and antimicrobial resistance of CRAB in bloodstream infections.

## 1. Introduction

*Acinetobacter baumannii*, a gram-negative pathogen, is a significant cause of hospital-acquired infections, including central line-associated bloodstream infections and ventilator-associated pneumonia [1,2]. *A. baumannii* infection is associated with increased mortality rates, as its high resistance to conventional treatments complicates the management of infections, potentially leading to an increase in fatality rates [2]. Carbapenem-resistant *A. baumannii* outbreaks often involve genetically similar strains, making multilocus sequence typing (MLST) a valuable tool for studying their spread. There are two primary MLST schemes for *A. baumannii*: the Oxford scheme, which distinguishes between closely related strains, and the Pasteur scheme, useful for studying the broader population structure and epidemiology of *A. baumannii* and related bacteria. In certain cases, the outcomes vary depending on the MLST type [1]. Carbapenems, crucial last-resort antibiotics, are effective against multidrug-resistant bacteria; however, the rise of carbapenem-resistant *A. baumannii* (CRAB) is a global concern. The World Health Organization has, since 2017, classified it as a priority 1 critical pathogen, particularly prevalent in intensive care units (ICUs) [3]. In Korea, ST191 was the most common CRAB type in ICUs from 2016 to 2017, with a notably high 30-day mortality rate [4]. Recently, ST369, showing stronger pathogenicity in mouse infection experiments, emerged. ST369 exhibits a higher incidence rate of CRAB bacteremia, a shorter time to bacteremia following ICU admission, and a greater early mortality rate, which may be attributed to its enhanced competitive growth rate and increased virulence compared to ST191 [1]. Despite the importance of epidemiological surveillance, genomic data on CRAB in Korea are lacking. Here, we report the draft genome sequence of diverse CRAB strains to elucidate its genomic features.

## 2. Results

### 2.1. Genome Assembly Quality

The genome sizes of the 12 CRAB strains ranged between 3,856,432 and 4,121,232 bp, with an average coverage of 524.207–707.341× and a GC content of 38.86–39.11%. The genome comprised 3621–3958 predicted protein-coding sequences, 72–74 tRNA genes, and 18 rRNA genes. Details of the assembly quality are provided in Appendix A.

### 2.2. Sequence Types and Phylogenetic Tree

Classification of the isolates by whole-genome multilocus sequence typing (wgMLST) yielded ST191, 195, 357, 369, 451, 469, 491, 784, 862, 1933, and 2929. G20AB08 was classified as a new sequence type (ST3326) based on novel combinations of the seven alleles (Appendix A).

The ANI tree analysis revealed differences among sequence types ST491, ST862, and ST2929, while the remaining types exhibited similar sequences (Figure 1). ST191 was found to be associated with ST469, 369, 195, 784, 491, and 3326.

### 2.3. Antimicrobial Resistance Genes

Identification of antimicrobial resistance determinants from the whole-genome sequencing (WGS) data revealed various antimicrobial resistance genes, predicting resistance to several antimicrobial classes, as shown in Figure 2. All the 12 CRAB isolates carried the *bla*_OXA-23_ gene, which is associated with carbapenem resistance. For aminoglycoside resistance, the *armA* gene was present in seven isolates; *aadA1* in six isolates; *aph(6)-Id* in five isolates; and *aph(3′)-Ia*, *aph(3′)*-Via, and *aac(3)-Ia* were each found in two isolates. Additionally, the *msr(E)* gene, linked to macrolide resistance, was found in six isolates, and the *tet(B)* gene, associated with tetracycline resistance, was present in five isolates. The *sul1* and *sul2* genes, which are related to sulfonamide resistance, were detected in seven and two isolates, respectively. Finally, the *catB8* gene, which may confer resistance to amphenicol, was observed in five isolates. Additionally, the *qacE* gene, which provides resistance to quaternary ammonium compounds used as disinfectants, was detected in seven isolates.

### 2.4. Virulence-Associated Genes

The investigation of virulence factors in all CRAB isolates identified genes associated with acinetobactin, trimeric autotransporters, biofilm formation, capsule production, HemO clusters, lipopolysaccharide, and outer membrane protein A (OmpA). Notably, variations in the biofilm, capsule, HemO cluster, and quorum sensing genes were observed across different *A. baumannii* sequence types. Details on virulence-associated genes and the results are presented in Figure 3.

## 3. Discussion

Within Asia, South Korea has been reported to have a relatively high proportion of drug-resistant *A. baumannii* strains, including CRAB and MDRAB (>93.0%) [5]. Therefore, we employed WGS to characterize the genomes of 12 CRAB isolates collected from hospitals in South Korea.

Among the collected isolates, differences were noted between sequence types ST491, ST862, and ST2929, whereas the remaining types showed similar sequences. ST191, the most prevalent sequence type reported in Korean ICUs and wards from 2016 to 2017 [4], was found to be closely related to several other isolates, including ST469, ST369, ST195, ST784, ST491, and ST3326. The newly classified ST3326 was identified as a subgroup of ST191, while ST357 was classified as a subgroup of ST3326.

All isolates exhibited characteristics of carbapenem resistance by harboring the *bla*_OXA-23_ gene, a carbapenemase gene that contributes to this resistance in the CRAB isolates. Similarly, the carbapenem resistance rate of *A. baumannii* strains isolated in Korea was recently reported to be 87.4%, with *bla*_OXA-23_ present in all strains [6]. According to a recent study, most (98.7%) of the CRAB isolates collected in Korea harbor the *bla*_OXA-23_ gene, while only a small number (1.3%) have the non-*bla*_OXA-23_ gene [7]. In this study, no isolates with the non-*bla*_OXA-23_ gene were found. Most of the isolates exhibited resistance to aminoglycosides. Aminoglycosides are frequently used as a treatment for CRAB, either alone or in combination with other antibiotics, such as polymixin and carbapenem [8,9,10]. *armA* (16S rRNA methyltransferase) was previously found in 80.2% of CRAB isolates in Korea [11]. This study also showed the presence of *armA* in most CRAB isolates (ST191, ST195, ST369, ST451, ST469, ST784, and ST3326).

Most isolates are susceptible to tigecycline (Appendix A), but tigecycline is not recommended for the treatment of CRAB infections [12,13]. Colistin is used as a last-resort antibiotic against CRAB, but only the ST369 strain among isolates was found to be resistant to colistin (Appendix A). *QacE*, which confers resistance to disinfectants via quaternary ammonium compounds, was found in ST369, ST191, ST491, ST784, ST469, ST2929, and ST3326. The differences in antibiotic resistance and disinfectant resistance genes may influence the spread of *A. baumannii* infections by overcoming infection prevention and control measures within the hospital environment.

Although the antibiotic and disinfectant resistance genes varied among the isolates, all domestic *A. baumannii* strains shared similar virulence factors, with the exception of genes related to biofilm formation and capsule production. The biofilm and capsule are well-established important virulence factors of *A. baumannii*. The ability to form biofilms contributes to the survival and transmissibility of *A. baumannii* in hospital environments, protects the microorganisms, and enhances resistance to various antibiotics [14,15]. Capsule formation significantly affects in vivo virulence by increasing resistance to desiccants, disinfectants, antimicrobials, and antibiotics [16]. Although detected in all isolates, *galU* and *galE*, which are associated with capsule production, are considered an important virulence factor in a number of Gram-negative pathogens [17,18,19]. Differences in these virulence factor-related genes, along with antibiotic resistance, may play a crucial role in the survival of *A. baumannii*. However, the pathogenicity of *A. baumannii* is influenced by various factors beyond the major virulence determinants. This is evident in the observation that, despite having the same major virulence factors, differences in virulence were observed between ST191 and ST369 [1]. Although these strains are phylogenetically similar, the ST369 strain is known to exhibit higher virulence and a faster growth rate than ST191 [1]. In this study, whole-genome comparison analysis revealed that the *OmpA* gene [20], a major virulence factor in *A. baumannii*, showed no differences between ST191 and ST369. Furthermore, the *wzc* gene [21], associated with the adhesiveness of *A. baumannii*, did not exhibit significant amino acid substitutions, such as G540A and G667D.

This study provides comparative genomic insights into bloodstream-infecting CRAB strains in Korea, serving as a foundation for future comparative genomic studies on the evolution and spread of this pathogen in human populations.

## 4. Materials and Methods

### 4.1. Strain Information

In 2020, CRAB strains were isolated from patient blood samples across nine general hospitals from the nationwide Korea Global AMR Surveillance System (Kor-GLASS). *A. baumannii* was isolated from blood using automated equipment, such as BaCT/ALERT (bioMérieux, Marcy l’Étoile, France). Bacterial testing from blood samples was routinely conducted for treatment purposes in ICUs and similar settings. After enrichment on tryptic soy agar at 37 °C (Becton Dickinson, Sparks, MD, USA), the strains were identified as *A. baumannii* using matrix-assisted laser desorption ionization time-of-flight mass spectrometry (bioMérieux, Marcy l’Étoile, France). In the 2020 Kor-GLASS project, 160 CRAB strains were classified into 12 MLST types, and 1 CRAB strain was randomly selected from each MLST type for WGS analysis.

### 4.2. Antimicrobial Susceptibility Testing

Antimicrobial susceptibility was assessed via the disk diffusion method, following the CLSI 2023 guidelines. The minimum inhibitory concentration values for imipenem and colistin were determined using the microplate dilution method with a Panel KRCDC2F (Thermo Fisher Scientific, TREK Diagnostic Systems, Cleveland, OH, USA).

### 4.3. WGS

The draft genome sequences of the 12 strains were obtained via combined analysis using the MiSeq system from Illumina and the MinION system with FLO-MIN114 from Oxford Nanopore Technologies at ONEOMICS (Gimpo-si, Gyeonggi-do, Korea).

### 4.4. Assembly, Annotation, and Characterization of Genome Sequences

Raw sequences were processed using trim_galore software v0.6.7 to collect, filter, and clean the data. De novo genome assembly was performed using the multi-platform genome assembly process (MpGAP) v3.1, using Nextflow version 21.10.6 and Masurca version 4.0.5. Gene prediction utilized ab initio methods, including Augustus v3.0, HiQ v1.0, GlimmerHMM v3.0.4, snap v2.5.5, and GeneMark2 v3.01.03. Annotation was conducted using InterProScan v5.63-95.0. Antimicrobial resistance and virulence genes were analyzed using the Center for Genomic Epidemiology ResFinder and virulence factor database (http://www.mgc.ac.cn/VFs/) accessed on 11 July 2024. The draft genome sequences of the 12 *A. baumannii* isolates were compared using wgMLST (https://github.com/bvalot/pyMLST) accessed on 5 July 2024, with ATCC 17978 serving as the reference strain.

## 5. Conclusions

In this study, we investigated the genetic diversity of CRAB strains linked to bloodstream infections in South Korea. The predominant sequence type, ST191, was found to have close genetic relationships with several other isolates, including ST469, ST369, ST195, ST784, ST491, and ST3326, with ST3326 classified as a subgroup of ST191. All examined strains harbored the *bla*_OXA-23_ gene, which is associated with carbapenem resistance. While variations in antibiotic and disinfectant resistance genes were observed, all isolates exhibited similar virulence factors. These results highlight the need for ongoing research to elucidate the evolution and transmission dynamics of this pathogen, informing effective therapeutic approaches and infection control strategies.

## Figures and Tables

**Figure 1 antibiotics-13-01124-f001:**
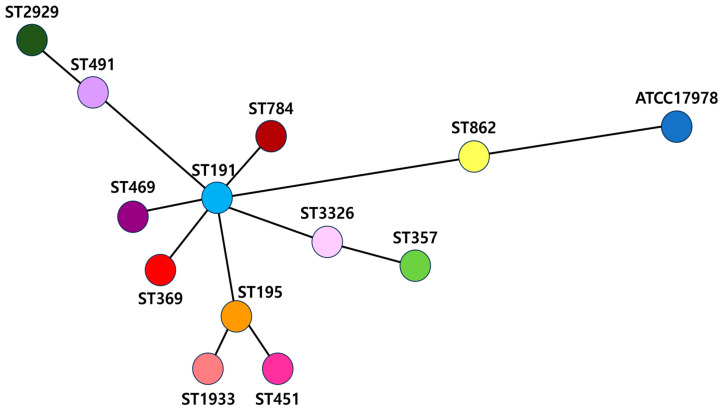
wgMLST (OXFORD)-based phylogeny of 12 *A. baumannii* isolates. *A. baumannii* isolates are depicted as circles interconnected by branches whose lengths correspond to the allelic distance.

**Figure 2 antibiotics-13-01124-f002:**
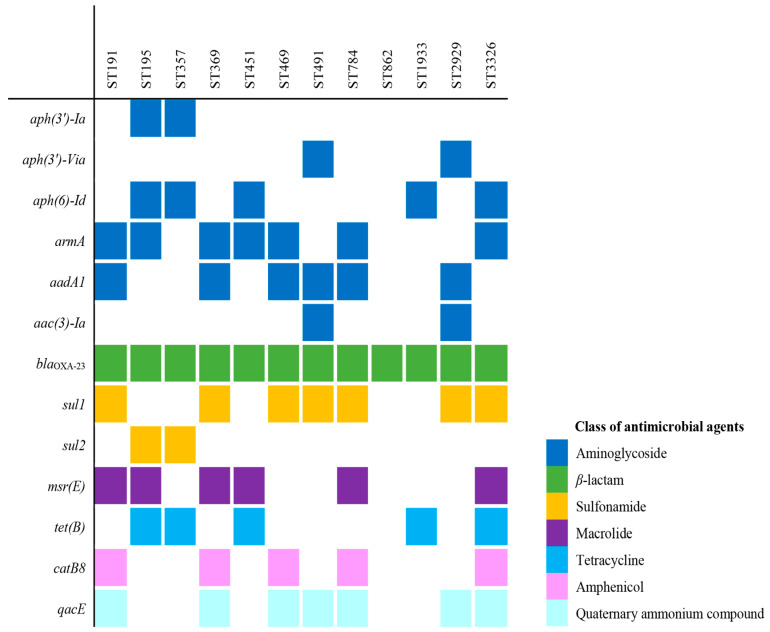
Antimicrobial resistance gene patterns in 12 carbapenem-resistant *A. baumannii* isolates from bloodstream infections.

**Figure 3 antibiotics-13-01124-f003:**
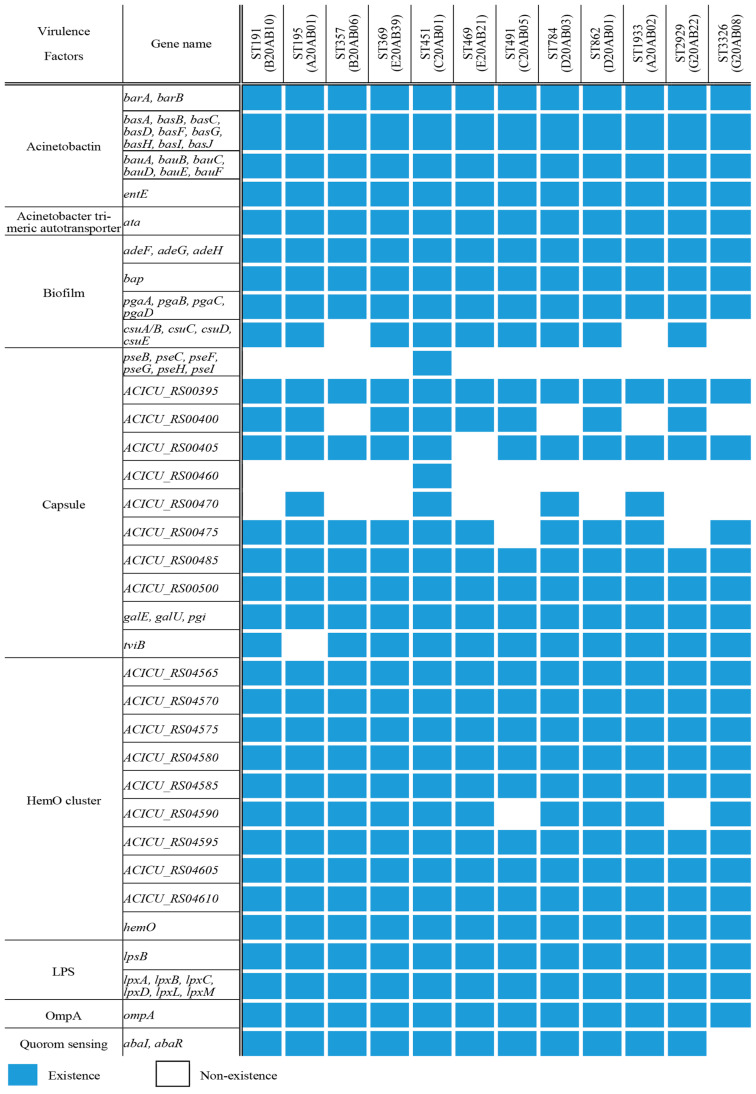
Pattern of virulence factors in the 12 carbapenem-resistant *A. baumannii* isolates from bloodstream infections.

## Data Availability

The complete genome sequence of isolates has been deposited in GenBank with the accession no. CP142895, CP143269, CP142101, CP142801, CP142102, CP142660, CP143262, CP142645, CP142642, CP145430, CP146812, and CP146231.

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
