# Peer review of "Genomic Analysis of Carbapenem-Resistant Acinetobacter baumannii Isolated from Bloodstream Infections in South Korea"

_antibiotics, 2024, doi:10.3390/antibiotics13121124_

Round 1
Reviewer 1 Report
Comments and Suggestions for Authors
In this short article, authors investigated clonal relationship and genetic features, of carbapenem-resistant Acinetobacter baumannii strains isolated from human blood infections in South Korea. The authors found large diversity in STs among 12 A. baumannii. The main carbapenem resistance gene was blaOXA-23. This is important information however the article needs revisions
Despite authors performed both short and long read, but the data presented here is merely descriptive. I think with hybrid assembly this study can determine in detail the genetic features on carbapenem resistance gene context etc..
Authors says predominant ST was ST191? How many isolates belonged to?
I suggest Authors can convert Table 1 and 2 into a single heatmap.
Authors have done hybrid assembly. If so, then plasmids description is missing, how many plasmids were found. Authors needs to determine which genes were located on plasmids (BRIG diagram or similar).
Genetic context that surronds blaOXA-23 should be analysed (Easyfig etc)
Author Response
The reviewer’s comments are indicated in bold and response by the authors are not in bold
Comments 1:
Despite authors performed both short and long read, but the data presented here is merely descriptive. I think with hybrid assembly this study can determine in detail the genetic features on carbapenem resistance gene context etc.
· This paper emphasizes the overall characteristics of the circulating strains of bloodstream infection-causing Acinetobacter baumannii in South Korea, categorized by their sequence types (ST). We used both short and long-read data to generate whole genome sequencing (WGS) results, which we plan to share with relevant researchers for further study. Key findings include an analysis of ST types through Average Nucleotide Identity (ANI) and insights into the resistance and virulence genes identified through WGS.
Authors says predominant ST was ST191? How many isolates belonged to?
· As indicated in the current paper, we found in 2020 that out of a total of 160 strains of carbapenem-resistant Acinetobacter baumannii (CRAB), 29 strains were classified as ST191. This makes ST191 the second most common type, following ST369, which had 57 strains. Additionally, the results from 2016 to 2020 indicate that ST191 accounted for 29% of all strains, representing the highest proportion.
I suggest Authors can convert Table 1 and 2 into a single heatmap.
· We have made the suggested changes (Figures 2, 3).
Authors have done hybrid assembly. If so, then plasmids description is missing, how many plasmids were found. Authors needs to determine which genes were located on plasmids (BRIG diagram or similar).
Genetic context that surronds blaOXA-23 should be analysed (Easyfig etc)
· We have already confirmed the structure surrounding the plasmid and blaOXA-23. The blaOXA-23 gene is located entirely on the chromosome, and no antibiotic resistance genes were found on the plasmid, indicating that there were no noteworthy findings to report in the paper. Please refer to the supplementary materials (Comment1-Supp.).
Reviewer 2 Report
Comments and Suggestions for Authors
The submitted manuscript provides important and critical information regarding Acinetobacter baumannii; however, I must mention a few major discrepancies in the information presented, namely:
Introduction Section: There is lacking information regarding the aspects presented in the manuscript, I strongly recommend improving this section
Line 23- please modify as blaOXA-48 as the standard notation for genes, and modify in all off the manuscript
Line 63- Regarding the results, there is no information about the disk diffusion method, whether if matches the genotypic results, or if the strains only present genes without expression. Please clarify.
Line 67-70- Please rephrase as the “…seven, six, five, two, two and two isolates…” as it is difficult to follow
Table 1- There are presented a large number of antibiotic classes, but must be mentioned the intrinsic resistance of A. baumannii
Moreover, there are discrepancies between disk diffusion method and genotypical results as presented in Table 1 and Table 3S. All the strains were presented susceptible to Minocycline, yet strains ST195, 357, 451, 1933, 3326 are presented with tet(b), which is expected to confer resistance to tetracyclines. Moreover, A. baumannii is presented in literature with intrinsic resistance. Please clarify.
Table 2- Please rearrange the information in the table, as some of the text is not visible. Additionally, clarify what authors aim to highlight by bolding the certain text, please describe in the text
Line 104- how was established the multidrug resistance, as there is no information described in the material and methods section
Line 116- The authors wrote “Most isolates are susceptible to tigecycline”. How authors established the susceptibility to tigecycline as CLSI 2023 does not recommend for testing, and there are not established the interpretation intervals
Material and methods- there is no information about the ethical approval or approval from an Ethics Board regarding the use of patients samples, please clarify
Line 148: There is no information on the method used to cultivate the samples. Did the authors use a classical method or an automated one? Please add to the manuscript
Line 149- It is unclear if sample identification and clinical isolates were processed only for this study or part of routine laboratory. Please add to the text
Line 151-In the results section there describes the use of an ATCC strain, which is not mentioned in the material and methods. Please clarify
Line 160-162 Why did authors perform MIC only for Imipenem and Colistin and not for Meropenem the standard screening method for carbapenemases? Using imipenem can provide false positive results for the production of carbapenemases. Please clarify
Author Response
The reviewer’s comments are indicated in bold and response by the authors are not in bold
Comments 2:
Line 22- please modify as blaOXA-48 as the standard notation for genes, and modify in all off the manuscript
· Thank you for your comments. We have made the indicated changes.
Line 63- Regarding the results, there is no information about the disk diffusion method, whether if matches the genotypic results, or if the strains only present genes without expression. Please clarify.
· The disk diffusion results are presented in Table S3 in the supplementary file.
· The results of the matching between genotype and phenotype are also shown in Table S3.
Line 67-70- Please rephrase as the “…seven, six, five, two, two and two isolates…” as it is difficult to follow
· We have revised the sentence.
Table 1- There are presented a large number of antibiotic classes, but must be mentioned the intrinsic resistance of A. baumannii
Moreover, there are discrepancies between disk diffusion method and genotypical results as presented in Table 1 and Table 3S. All the strains were presented susceptible to Minocycline, yet strains ST195, 357, 451, 1933, 3326 are presented with tet(b), which is expected to confer resistance to tetracyclines. Moreover, A. baumannii is presented in literature with intrinsic resistance. Please clarify.
· According to a review paper titled "Acinetobacter baumannii Antibiotic Resistance Mechanisms," A. baumannii exhibits intrinsic resistance to antibiotics, such as penicillins and cephalosporins.
· Additionally, genotype and phenotype results do not always correlate. A recent study (titled "Combinations of Antibiotics Effective against Extensively- and Pandrug-Resistant Acinetobacter baumannii Patient Isolates") reported cases where the presence of the tetB gene did not confer resistance to minocycline.
· Therefore, in this paper, we modified Table 1 to Figure 2 to emphasize the status of resistance genes in A. baumannii strains of different sequence types.
Table 2- Please rearrange the information in the table, as some of the text is not visible. Additionally, clarify what authors aim to highlight by bolding the certain text, please describe in the text
· We have revised Table 2 for clarity. Please refer to Figure 3.
Line 104- how was established the multidrug resistance, as there is no information described in the material and methods section
· We have revised the content of the main text to address this.
Line 116- The authors wrote “Most isolates are susceptible to tigecycline”. How authors established the susceptibility to tigecycline as CLSI 2023 does not recommend for testing, and there are not established the interpretation intervals
· The relevant breakpoints have been noted in Supplementary Table S3.
Material and methods- there is no information about the ethical approval or approval from an Ethics Board regarding the use of patient’s samples, please clarify
· We have included the ethical statement in the main text.
Line 148: There is no information on the method used to cultivate the samples. Did the authors use a classical method or an automated one? Please add to the manuscript
· We have described the use of an automated equipment for this purpose in the materials and methods section.
Line 149- It is unclear if sample identification and clinical isolates were processed only for this study or part of routine laboratory. Please add to the text
· We have noted in the main text that bacterial identification and susceptibility testing from blood samples are conducted repeatedly in the hospital.
Line 151-In the results section there describes the use of an ATCC strain, which is not mentioned in the material and methods. Please clarify
· We have indicated this in the materials and methods section.
Line 160-162 Why did authors perform MIC only for Imipenem and Colistin and not for Meropenem the standard screening method for carbapenemases? Using imipenem can provide false positive results for the production of carbapenemases. Please clarify
· There were results for meropenem in the supplementary data, but some were not visible. We have adjusted the page size to ensure that the meropenem results are now displayed properly.
Reviewer 3 Report
Comments and Suggestions for Authors
The article entitled "Genomic Analysis of Carbapenem-Resistant Acinetobacter baumannii Isolated from Bloodstream Infections in South Korea" offers an interesting and fresh perspective on the Carbapenem-Resistant strains of Acinetobacter baumannii isolated from blood cultures. The article is well written and should be considered for publication, as this is a subject of interest and a serious matter of concern worldwide.
However, I would like to suggest some improvements:
- line 32 - please modify "A. baumannii" to "Acinetobacter baumannii". I would consider this to be more suitable, at least the first time the bacteria is mentioned in the article
- please remake the graph in Figure 1 - the text is sometimes difficult to read
- please modify the paragraph from lines 64-76 to be a little more comprehensive. It is difficult to follow which genes were present in how many of the studied isolates
- please modify the text size and arrangement in Tables 1 and 2. Due to the small dimensions of the text, they are very hard to follow
- furthermore, for both tables but especially for Table 2, I think the tables would benefit from an explanation of the gene names/purpose in the table caption in order to make the information more easily accessible
- please also rearrange the tables in the Supplementary materials to be more easy to read and understand
- in the Materials and methods section, could the authors please clarify why the 12 strains were chosen randomly and the reason why the study was limited to only 12 strains for whole-genome sequencing?
Comments on the Quality of English LanguageEnglish language could use some minor improvements.
Author Response
The reviewer’s comments are indicated in bold and response by the authors are not in bold
Comments 3:
Line 32 - please modify "A. baumannii" to "Acinetobacter baumannii". I would consider this to be more suitable, at least the first time the bacteria is mentioned in the article
· We have made changes to address this.
Please remake the graph in Figure 1 - the text is sometimes difficult to read
· We have revised the ST numbers for readability.
Please modify the paragraph from lines 64-76 to be a little more comprehensive. It is difficult to follow which genes were present in how many of the studied isolates
· We have revised the sentences for clarity.
Please modify the text size and arrangement in Tables 1 and 2. Due to the small dimensions of the text, they are very hard to follow
· We have revised the presentation of Tables 1 and 2.
Furthermore, for both tables but especially for Table 2, I think the tables would benefit from an explanation of the gene names/purpose in the table caption in order to make the information more easily accessible
· We have revised the presentation of Table 2.
Please also rearrange the tables in the Supplementary materials to be more easy to read and understand
· We have revised the format of the tables in the supplementary materials.
In the Materials and methods section, could the authors please clarify why the 12 strains were chosen randomly and the reason why the study was limited to only 12 strains for whole-genome sequencing?
· We have indicated this in the materials and methods.
Round 2
Reviewer 2 Report
Comments and Suggestions for Authors
Thank you for youre kind response
Author Response
Comment 2: Thank you for youre kind response
response: Thank you.
Reviewer 3 Report
Comments and Suggestions for Authors
I agree with the changes made during the first round of review and I consider that the article is suitable for publishing.
My only suggestion would be to revise Figure 2 before giving the final decision, as I think some of the columns are not properly fitted (there are strains that have no color on their column, but at the end on the right side of the table, there are some columns with no strain numbers that contain some colors under them, which makes me believe that the columns were accidently shifted when exporting the figure).
Author Response
The reviewer’s comments are presented in bold font and responses by the authors in regular font.
Comments 3:
I agree with the changes made during the first round of review and I consider that the article is suitable for publishing.
My only suggestion would be to revise Figure 2 before giving the final decision, as I think some of the columns are not properly fitted (there are strains that have no color on their column, but at the end on the right side of the table, there are some columns with no strain numbers that contain some colors under them, which makes me believe that the columns were accidently shifted when exporting the figure).
· Thank you very much for your review. I have revised Figures 2 and 3 in the manuscript.